# Understanding Surface Modulation to Improve the Photo/Electrocatalysts for Water Oxidation/Reduction

**DOI:** 10.3390/molecules25081965

**Published:** 2020-04-23

**Authors:** Yunhee Cho, Thi Anh Le, Hyoyoung Lee

**Affiliations:** 1Center for Integrated Nanostructure Physics (CINAP), Institute for Basic Science (IBS), Sungkyunkwan University, Suwon 16419, Korea; cyhee94@gmail.com (Y.C.); leanhmse@gmail.com (T.A.L.); 2Department of Chemistry, Sungkyunkwan University, Suwon 16419, Korea; 3Department of Biophysics, Sungkyunkwan University, Suwon 16419, Korea

**Keywords:** photo/electrochemical catalysts, interface modification, surface functionalization, water oxidation, water reduction

## Abstract

Water oxidation and reduction reactions play vital roles in highly efficient hydrogen production conducted by an electrolyzer, in which the enhanced efficiency of the system is apparently accompanied by the development of active electrocatalysts. Solar energy, a sustainable and clean energy source, can supply the kinetic energy to increase the rates of catalytic reactions. In this regard, understanding of the underlying fundamental mechanisms of the photo/electrochemical process is critical for future development. Combining light-absorbing materials with catalysts has become essential to maximizing the efficiency of hydrogen production. To fabricate an efficient absorber-catalysts system, it is imperative to fully understand the vital role of surface/interface modulation for enhanced charge transfer/separation and catalytic activity for a specific reaction. The electronic and chemical structures at the interface are directly correlated to charge carrier movements and subsequent chemical adsorption and reaction of the reactants. Therefore, rational surface modulation can indeed enhance the catalytic efficiency by preventing charge recombination and prompting transfer, increasing the reactant concentration, and ultimately boosting the catalytic reaction. Herein, the authors review recent progress on the surface modification of nanomaterials as photo/electrochemical catalysts for water reduction and oxidation, considering two successive photogenerated charge transfer/separation and catalytic chemical reactions. It is expected that this review paper will be helpful for the future development of photo/electrocatalysts.

## 1. Introduction

The utilization of fossil energy produces various pollutants, increases atmospheric CO_2_ concentration, and causes further green-house effects and climate change. Solar energy is one promising, abundant alternative energy source that can reduce air pollutants and minimize energy costs. There are various ways to utilize solar energy, including converting solar energy into heat energy, electrical potential energy, and chemical energy. Due to the daily and seasonal variability of solar energy, energy conversion into chemical energy, in the form of valuable fuels such as H_2_, CH_4_, and alcohol, is an effective way to utilize solar energy. However, the energy conversion efficiency of this route is still very low because most of the chemical reactions for the production of fuels are endothermic reactions, which require a large energy to overcome the difference in Gibbs free energy between the reactants and products [1,2]. Various catalysts can reduce the energy barrier for these reactions, control the reaction mechanism and reduce unwanted byproducts.

Two processes, including the charge transfer/separation and successive catalytic chemical reaction at the surface of the materials, are critical for high energy conversion efficiency. When a semiconductor absorbs light with energy that is larger than the material’s band gap, carriers are excited, and excitons, that are coupled charge carriers by Coulomb forces, are generated. The excited carriers obtain potential energy from solar energy and give the kinetic energy to the surface-adsorbed electron and/or hole acceptors to overcome the activation energy for the generation of valuable fuels. The various chemical conformations of the chemical reactants (e.g., H_2_O, H_3_O^+^, or OH^−^) which is determined by the electrolyte conditions, such as pH, change the energetic states of adsorption/desorption of reactants/products. Thus, maximization of photocatalytic efficiency requires understanding how the semiconductor (SC) absorbs and generates excitons (Equation (1)), how the generated excitons transfer along the semiconductors (Equation (2)) and how the electron/hole acceptors are reduced/oxidized on the electrode (Equations (3)–(8), Equations (5)–(8), * refers to the adsorption of the species on the active sites of the catalysts) [3]. However, catalysts that exhibit both adequate light absorption and efficient catalytic activity are rare. For example, Si is well known for its ability to absorb light from a broad wavelength range. In contrast, the limited chemical stability of Si under the acidic and basic catalytic operation condition and its low catalytic activity are problematic for solar energy conversion capabilities [4]. Therefore, combining catalysts with light-absorbers is required to achieve the proper reproducibility and catalytic efficiency. Besides, understanding the charge transfer/separation between absorber-catalysts (solid-solid) and catalysts-reactants (solid-liquid) and catalytic reaction between catalysts-electrolyte becomes significant for high photo-energy conversion efficiency. We first consider the photogenerated charge carrier transfer and separation process and consider the catalytic chemical reaction on the surface subsequently.
SC + h*v* → SC + (e^–^ + h^+^)_bulk_(1)
(e^−^ + h^+^)_bulk_ → (e^−^ + h^+^)_surface_(2)
2H^+^ + 2e^−^ → H_2_ E^0^ (vs NHE) = −0.41 V(3)
H_2_O → 1/O_2_ + 2H^+^ + 2e^−^ E^0^ (vs NHE) = 0.82 V(4)
OH^−^ → OH* + e^−^(5)
OH* + OH^−^→ O* + (H_2_O + e^−^)(6)
O* + OH^−^ → OOH* + e^−^(7)
OOH* + OH^−^ → O_2_ + H_2_O + e^−^(8)

### 1.1. Photogenerated Charge Carriers Dynamics

The excited charge carriers are generated by incident light, transfer and separate into the electron/hole acceptors at the surface of an electrode (Figure 1). Once the excitons are generated by the light absorption (Figure 1 (i)), the photogenerated carriers transfer toward the surface of the material according to the charge carrier concentration gradients and/or the applied bias (Figure 1 (ii)). However, if the charge carriers cannot transfer rapidly due to intrinsic electrical conductivity limitations of the materials, the excitons relax to the ground state and generate heat or light (Figure 1(iii)). Additionally, charge carriers can be trapped in trap states, such as antisites, interstitials, and vacancy defects (Figure 1 (iv)) [5,6,7,8,9]. The surface contains a high density of vacancy defects because the bulk crystalline structure does not elongate further. The disconnectivity of the crystalline structure generates various dangling bonds and creates unpaired electrons, surface dipoles, and trap states, which are located away from the minimum conduction and maximum valence band edges. Surface defects affect the Fermi level at the surface of the materials and change the concentration of the charge accumulated at the surface. Besides, the generated charge carriers are trapped at these defect states on the surface. This phenomenon reduces the concentration of available charge carriers, which are consumed in catalytic reactions on the surface, and decreases the energy conversion efficiency.

Following the accumulation of charge carriers on the surface, charge carriers transfer into the chemical reactants in the liquid electrolyte. Because of the mismatch of the Fermi levels between the semiconductor and the electrolyte solution, the charge concentration in the semiconductor is redistributed to establish an equilibrium at the interface. This change results in a shift of the Fermi level and generates the band bending at the interface (Figure 1(v)). Subsequently, upward band bending in the semiconductors enhances the photovoltage that provides extra kinetic energy for the catalytic reaction of electron acceptors (Figure 1 (vi)). In particular, the nanomaterials require the efficient band bending modulation for charge separation because charge carriers are easily recombined due to their highly localized wavefunctions from size-confinement effects. The band bending, in other words, the built-in potential inside the materials could accelerate the charge separation into the electron/hole acceptors [10,11]. The band bending is dependent on the charge carrier concentration, and the depletion length [12]. Fortunately, the surface structures of nanomaterials can modulate the charge carrier densities in the materials significantly due to the large surface-to-volume ratio to increase the band bending [13,14]. Besides, the depletion length which is the distance from the surface to the change neutral region in the materials, is strongly dependent on the surface structures [15,16]. Thus, various chemical surface modulation methods have been applied further to control the direction and magnitude of the band bending.

The surface-accumulated photogenerated electrons/holes are consumed in the chemical reaction. Unfortunately, only one reduction or oxidation reaction can be preferred due to the slow kinetics of the one-half reaction or a mismatch of the energy levels of a semiconductor and the orbital levels of electron and/or hole acceptors. As result, the imbalanced accumulation of charge carriers in the semiconductor recombines with excited charge carriers to reduce the charge extraction efficiency. The accumulation of charge carriers on the surface also causes photoreduction/oxidation of the materials themselves, which is known as photo-corrosion. Photocorrosion degrades the active catalytic sites on the surface and reduces the long-term reproducibility of the materials. So, understanding and balancing the coupled reduction and oxidation reaction on the surface becomes critically essential for the ability to retain the long-term catalytic response and to enhance the photo-energy conversion efficiency.

### 1.2. Catalytic Chemical Reaction: Water Reduction/Oxidation Reaction

Generally, the essential descriptors also should be differentiated because the electrocatalytic mechanism of HER/OER is different distinctively. Below, we provide a summary of the current efforts aimed at tailoring the critical factors of electronic structure and surface/interface geometry for the two essential reactions of the water electrolysis reaction, water reduction (HER), and water oxidation (OER), for hydrogen production. Recent research efforts have sought to achieve improved catalytic performance by leveraging such vital factors.

Briefly, in an electrolyzer cell, the hydrogen evolution reaction (HER) takes part in the cathode with the involvement of two transferred electrons (Equation (3)). In the electrochemistry literatures, it is widely accepted that HER performance is strongly sensitive to the pH of the electrolyte; it becomes more sluggish when operated under basic conditions compared to acidic conditions. Up to now, literatures have recorded the large number of experiments and theoretical reports in water reduction reaction under acid electrolytes [17]. However, for the sake of cost-effective technology in practical application, alkaline aqueous solution is more usually a good choice for hydrogen source. Under basic condition, in the very beginning step, water molecules have to be dissociated into H* and OH*, which then further react with other intermediates through proton-coupled electron transfer to form the final products [18]. To be more specific, for instance, under the high pH electrolyte, the massive of hydronium ions are locally accumulated at the surface of catalysts during water dissociation and hydrogen adsorption steps, which then generates the acidic reactive environment driven by the H_3_O^+^ intermediates [18,19,20]. As consequence, the energy barrier of the overall reaction was decreased, leading to higher HER performance. Recent studies about the non-noble electrocatalyst based on transition metals have proved that, the incorporation of an early transition metals (i.e., Ni, Co) can dramatically boost the water dissociation step ascribed to the favorable interaction of surface-adsorbed OH_ads_ with relevant intermediates [21,22,23]. As in above-mentioned, the binding strength of adsorbed hydrogen is a key determinant of the concentration of the adsorbed hydrogen intermediate on the surface. In short, the HER kinetics commonly decrease with increase of the pH. Moreover, depending on the pH of the electrolyte solution, different chemical reactants, for example, hydronium ions (H_3_O^+^_,_ in acidic condition) or water molecules (H_2_O, in basic condition), diffuse and adsorb on the catalysts by accepting an electron to form an adsorbed hydrogen intermediate. It is important to note that, the variations in activity of an electrocatalyst for HER reaction are closely related to the hydrogen adsorption energies. This concentration governs the mechanism of HER: it should neither be too strong nor too weak to facilitate the reaction. When coverage of adsorbed hydrogen is high, the nearby hydrogen intermediates react and generate hydrogen, H_2_, a process known as the Volmer-Tafel mechanism. However, in case of low hydrogen coverage, hydrogen intermediates react with protons by supplying an electron to generate hydrogen, a process known as the Volmer-Heyrovsky mechanism. So, the surface chemical properties that affect the concentration of reactants at the interface are essential to the catalytic reaction process [3,24].

On the anodic side, oxygen evolution reactions (OER) take place, which are more complicated than the HER due to the involvement of four electrons. It is widely accepted that the OER in basic media is more favorable than that in an acidic electrolyte, due to the high availability of OH^−^* intermediates in alkaline solution (Equations (5)–(8)). According to the pH condition of the electrolyte, the chemical species change from hydroxyl ions (in basic) to water molecules (in acidic). Besides, intermediates and byproducts, are different under acidic and basic conditions, indicating that the surface chemical mechanism depends on the pH. Usually, OER in acidic conditions has difficulties in improving the total conversion efficiency. In particular, the high Gibbs free energy of the formation of OOH intermediates with proton-coupled electron transfer process or that of the double bond formation for oxygen molecules should be overcome [25]. Thus, a catalytic reaction requires appropriate selection and modification of catalysts to control the complex physical and chemical reactions in the photocatalysts [26,27,28,29].

In particular, for the process of solar energy conversion into chemical energy via photocatalysis, there are two kinds of reactants in the catalytic reaction: charge carriers including electrons and holes, and chemical species, such as protons, water molecules and hydroxyl anions. To maximize the photocatalytic performance, (1) photo-generated charge carriers should transfer and accumulate on the surface for separation into adsorbed reactants and (2) adequate amounts of the chemical species should be adsorbed on the surface to react with charge carriers. Each of these processes is strongly affected by both the electronic and chemical structure of the solid-solid and solid-liquid interface. The surfaces of nanomaterials have different electronic and chemical structures than the corresponding bulk materials, containing various surface vacancy defects and charged dangling bonds due to the discontinuous crystalline structure. Thus, the energetic diagram of the catalyst is strongly affected by the surface structure that changes the flow of photo-generated carriers. Herein, we describe strategies to enhance the charge transfer/separation and the surface chemical reaction, which are expected to be great guidelines for the rational design of future photo/electrocatalysts.

First, we consider the surface structure of materials resulting from an intrinsic surface-terminated chemical structure, such as surface defects, crystalline facets with different coordination states, and passivation with atomic elements. The interface properties can be modulated through functionalization with various chemical reagents. We describe the effects of the intrinsic dipole moment of chemical reagents and chemical interaction with the electrode materials at the solid-liquid interface. These methods control the charge transfer and separation in the electrode and consequently change electrochemical properties, such as the photovoltage, barrier height, and overpotential for the catalytic reaction. The chemical modification of the interface results in modulation of the concentration of accumulated charge carriers and the potential energy of charge carriers at the solid-liquid interface.

Second, we describe the strategies for increasing electrocatalytic activities by modulating the surface of the materials. The precise control of the electronic and chemical structure of the surface can enhance the reactant absorption and the catalytic reaction kinetics. In particular, the surface chemical structure can tailor the chemical potentials for adsorbing chemical reactants, such as protons, water molecules, and hydroxyl ions on the surface and control the Gibbs free energy of intermediate states during the catalytic process. The material surface can be modulated by controlling the surface-terminated structure and introducing chemical functionalities.

## 2. Surface Modulation for Improved Charge Transfer and Separation in the Photocatalytic Process

The surface of a photoelectrode is different from the bulk crystalline structure of a catalyst. The surface structure of the electrode is strongly correlated with the charge transfer/separation at the electrode-electrolyte interface. At the solid-liquid interface, free charge carriers flow between the surface of the photoelectrode and the electrolyte solution until they reach an equilibrium state. This flow is due to the difference between the electrode and electrolyte Fermi levels. In the case of a cathodic photoelectrode, the band structure at the interface shifts negatively, which is known as band pinning, and the concentration of the majority carrier is depleted [30]. As a result, the built-in potential at the surface leads to charge accumulation on the surface, which can enhance the charge separation to the surface and supply the large kinetic energy for the catalytic reaction. The magnitude and the direction of the surface dipole can be modulated by control of the surface structure, including surface-terminated chemical structure, surface functionalization with molecules, and introduction of buried junctions and adaptive junctions to control and enhance the charge transfer process for the photo- and photoelectro-catalytic reaction at the interface.

### 2.1. Surface States at the Solid-Liquid Interface for Charge Transfer and Separation

The surface chemical structure of materials has different chemical and electronic properties from the bulk crystalline structure, which generates charged dangling bonds, various coordination states, and different chemical potential. The charged dangling bonds on the surface induce a surface dipole, and induce the Fermi level shift at the surface to produce a built-in potential [7,8]. Also, the facets of the surface have distinctive atomic arrangements, and coordination states, and possess the different chemical and electronic properties. As a result, facets on the surface induce anisotropic charge transport and chemical activity towards the water reduction and oxidation reaction [31]. Thus, understanding and modifying the surface chemical structures at the solid-liquid interface is a good starting point for enhancing solar energy conversion efficiency.

Surface chemical structures have a high density of vacancy states and generate charged dangling bonds. These result in trap states, extra charge carriers, and surface dipoles, which cause the difference of Fermi level. Also, a built-in potential is generated due to the difference in the Fermi level compared to the Fermi level of the electrolyte solution. The band alignments at the surface affect the concentration of accumulated photo-induced charge carriers on the surface and supply additional potential energy for the catalytic reaction on the surface. For example, in TiO_2_, a well-known photocatalyst, the oxygen vacancy states generate interstitial Ti^3+^ or Ti^3+^ states on the surface [14,32,33]. These chemical species induce shallow trap states below the conduction band [34,35]. The metal vacancies in TiO_2_ generate trap states near the valence band to broaden the band and increase the hole mobility [36]. Abdullah et al [34] confirmed that charge carriers accumulate in the trap states and proceed to radiative recombination. In the one-element semiconductor of a diamond, which consists of sp^3^-hybridization, the surface has a high electric dipole moment when it is terminated with hydrogen, which leads to an upward shift of the surface band edge. This shift causes the accumulation of the holes on the surface, resulting in higher p-type conductivity and easier oxidation of the adsorbed water molecules. When a diamond is terminated with oxygen, which is more electronegative than carbon, the band is bent downward [37,38]. Surface termination in materials of two or three-element also changes the surface dipole, Fermi level at the surface, and overpotentials for the catalytic reaction [39,40,41,42].

The surface structure in accordance to the crystalline facets, which have the ordered atomic arrangements and distinctive coordination states have characteristic electronic and chemical properties. These atomic arrangements also change the charge carrier accumulation and chemical potential at the surface. Considering a single crystal, the anisotropic charge transfer depending on the surface facets were presented because of the Fermi level and band bending [43]. For example, the anatase TiO_2_ crystalline structure presented the facet-sensitive Fermi level between the (001) and (101) facets, as observed by X-ray photoelectron spectroscopy and ultraviolet photoelectron spectroscopy. Fine control of surface termination by sputtering, annealing, and reoxidation confirmed that the higher densities of deep trap states such Ti^3+^ interstitials and oxygen vacancies were generated dominantly in (001) facets. In addition, the variation of the Fermi level of exposed facets is larger than that of the (101) facets, while the Fermi level of the (001) facets is always higher than that of the (101) facets [44]. Surface dipoles also depend on the facets, which also change the barrier height at the surface (Figure 2) [44]. The (001) facets of anatase and the (110) facets of rutile TiO_2_ have been analyzed with time-resolved photoelectron spectroscopy to confirm the carrier lifetime and the band bending when irradiated with light (Figure 2). The magnitude of the negative band bending is larger in anatase (001), whereas the complete charge carrier relaxation time is lower for (110) rutile TiO_2_. This comparison indicates that the concentration of transferred and accumulated electrons on the surface is higher in the case of (001) facets of TiO_2_. However, the surface defect concentration on the surface affects the barrier height. And the higher barrier height of the (110) rutile facets increases the photoexcited electron relaxation time. The exposed facets change the Fermi level, the band bending, and the barrier height at the interface and affects the charge transfer and separate process. Thus, the photo- and photoelectrocatalytic reaction is dependent on the surface exposed facets. For example, in SrTiO_3_, the photogenerated hole is preferred to be accumulated on the (110) facets to oxidize the adsorbed chemical species, while the (001) facets prefer electron extraction. This difference results from exposed elements on the surface inducing the surface dipoles, which leads to specific charge carrier transfer and accumulation on the surface [40,45].

The surface dipole induced by the surface dangling bonds can be modulated by surface passivation with other atomic species [46,47,48,49]. For example, when the O-polar surface of single-crystal ZnO is passivated with hydrogen, the electronic band shifts from downward to upward with increased coverage of hydrogen due to the electron donation of hydrogen (Figure 3) [46]. Also, when the oxygen-defected TiO_2_ is functionalized with fluorine, the highly electronegative fluorine withdraws the electron densities from Ti^3+^ nearby [47]. Metal incorporation onto the surface is an effective way to control the overpotentials for the reaction [50,51,52] is because the frontier d orbitals of metal cation directly relates to the edge of the band structure and changes the surface polarization by modulation of densities of charge carriers and the position of band edges [50,53,54]. For example, as the aliovalent Sr^2+^ cation is substituted the La^3+^ in the LaFeO_3_ epitaxial film, the doped metal the difference in valence band edge and the Fermi level. The modulation of the band edge resulted in the reduction of the photovoltage and enhanced the water oxidation performance by rapid and strong binding with the water as reactants [51]. And in case of SrTiO_3_, Rh and Ni states improve the extraction of electrons and holes at the same time, whereas the other dopants such as Cu, Fe and Mn only allow the electrons to extract for the hydrogen evolution by changing the photovoltage [52].

Surface structures such as defects, facets, and passivation with elements, change the Fermi level of the surface. The charged dangling bonds on the surface; in other words, surface defects induce surface dipoles, change the band edge shift, and the charge accumulation on the surface. Also, the coordination state on the surface strongly affects the chemical properties of the elements. As a result, if the Fermi level is up-shifted in the surface of the cathode, the overpotential increases to supply enough energy for the catalytic reaction on the surface [47]. However, the high densities of surface defect states act as trap states on the surface that accept the charge carriers and recombine with the transferred charge carriers on the surface, degrading the efficiency of the device. So, methods to regulate the densities of surface defects, such as introducing passivation layers, oxygen plasma exposure, hydrogen treatments and annealing are required [9,55,56,57,58]. In addition, we cover the methods for regulating the type of defects and densities of defects in Section 3 in details.

### 2.2. Surface Functionalization at the Solid-Liquid Interface for Charge Transfer and Separation

Functionalization of the surface with molecules causes electron density redistribution between the substrate and the adsorbed molecules. The intrinsic dipole of the molecules and the electronic interaction between molecules and the substrate can influence the polarization at the interface to change the band bending on the surface. [12,30,59,60]. Surface functionalities tune the magnitude and direction of the surface dipole. Thus, controlling surface band edge positions can be achieved by surface functionalization. In previous reports, various functionalities on the surfaces of materials, in particular the molecular functionalization, have been shown to control the surface dipole at the interface effectively and modulate the band edge position of nanomaterials to align relative to the water oxidation/reduction potentials [61,62,63,64]. Also, the creation of the band bending on the surface improves the barrier height at the interface between the electrode and the electrolyte solution, which is similar to a Schottky junction of a semiconductor-metal junction [65]. The magnitude and direction of the energy barrier control the type of charge carrier accumulation on the surface and enhance the separation of the coupled charge carriers. Thus, surface functionalities change the electronic structure at the solid-liquid interface to modulate the charge transfer and separation.

The anchoring groups of the incorporated organic molecules change the surface charge transfer resistance. When a p-Si/TiO_2_ photoelectrode was functionalized with molecular catalysts with different anchoring groups, including carboxylate, phosphonate, and hydroxamate, the device that is functionalized by hydroxamate groups showed the lowest charge transfer resistance and highest short-circuit current. This result indicates that chemical interaction with the anchoring groups of the molecules redistributes electron densities on the surface and tunes the accumulation of charge carriers and separation into the reactants on the surface [66]. Additionally, the intrinsic dipoles of the functionalities can change the redistribution of the electron densities [67,68,69,70]. For example, the Ag nanoparticles functionalized with electron-withdrawing thiocyanate (SCN-) ligands, enhance the electron accumulation and transfer from Ag to proton as electron acceptors effectively [71]. However, Pt- or Au-loaded TiO_2_ showed a negative shift of overpotential for HER, implying that the exposed metal sites are the active sites instead of the adsorbed ligands, and the surface functionalities modulate the accumulated charge carrier densities on the metal sites [68,71]. Furthermore, the surface dipole at the solid-liquid interface can be modulated by the intrinsic dipole moments, the geometrics and orientation of ligands functionalized on material, and the interaction between adjacent molecules in different nanomaterials can enhance the band edge modulation over 2.0 eV [72,73].

The adequate alignment of band levels of catalysts compared to the redox potential of water molecules at the interface is import for the transfer of charge carriers to the electron/hole acceptors. Surface dipoles which are induced by surface functionalities on the surface modulate the band bending at the interface, which affects the degree of orbital overlapping between the catalysts and the charge carrier acceptors for the efficient catalytic reaction. For example, three-atomic layer nanosheets of MoS_2_ are known as promising HER catalysts due to their relatively low required free energy for hydrogen adsorption compared to other materials [74,75,76]. To achieve the proper energy alignment for efficient charge separation into the adsorbed proton, the MoS_2_ was functionalized with aromatic molecules with various substituents and the Fermi levels were rearranged according to the electronegativities of the molecules due to the electron density redistribution [77,78]. Additionally, the coverage of the ligands on the materials affects the magnitude of the band edge shift without significant band gap change [79]. Thus, when the octahedral structure of the MoS_2_ nanosheets was functionalized with organic molecules with electron-donating groups, the HER performance was improved, and stability was increased due to charge redistribution between the MoS_2_ and the functionalized molecules. In addition, the metal oxide MnO catalysts on BiVO_4_/WO_3_ functionalized with BF_4_^−^ oxidizes water molecules under neutral condition, showed the enhanced catalytic performance. This result is due to the band edge modulation created by ligand functionalization on the surface [80]. The barrier height at the solid-liquid interface, which is measured as open-circuit voltage, can be controlled by the surface functionalities [73,81]. The surface of p-Si was functionalized with CH_3_, p-nitrophenyl, and m-dinitrophenyl molecules. As a result, the barrier height at the solid-liquid interface increased as the electrostatic surface dipoles negatively shifted the valence band. Equilibration of charge carrier concentration between the photoelectrode and the electrolyte solution creates the band bending, which modulates the barrier height and the photovoltage [81].

In conclusion, the energetic surface states of various materials, including metal oxides, metal chalcogenides, and metal nanoparticles, can be modulated by introducing a surface dipole that is induced by functionalization of the surface with molecules. The magnitude and direction of the surface dipole are controlled by the electronic interaction between molecules and the substrates, and the intrinsic dipole moments of the molecules. As a result, the accumulation of charge carriers at the surface and charge transfer kinetics to the adsorbed electron/hole acceptors are controllable with various surface functionalizations. In addition, considering the surface-to-volume ratio, the band edge of materials with smaller dimensions, including zero-dimensional quantum dots and quasi-two-dimensional nanoplatelets, are strongly modulated by functionalization with molecular ligands [82,83,84]. Thus, surface functionalities on the nanostructured electrode materials become important for controlling the energetic states at the surface.

### 2.3. Modulation of Solid-Solid Interface for Charge Transfer and Separation

One method of achieving energy conversion efficiency is to use a multi-junction in the electrode, such as coupling of the catalysts on the light-absorbing semiconductor surface, or an adaptive junction to separate the charge carriers efficiently. Functionalization of the electrode with molecules suffers from the limited stability during catalytic operation under harsh condition, and compensation of the surface dipole of functionalizing molecules that is caused by the electrolyte double layer [85,86,87]. Thus, controlling the solid-solid junction is another way to control the charge transfer and separation processes, which determines electrochemical properties such as the generated built-in potential, open-circuit potential, and photovoltage [1,88,89]. Insertion of different materials inside the electrode induces a new charge concentration equilibrium at the solid-solid junction and induces band bending that controls the charge transfer and accumulation at the solid-liquid interface. In addition, the insertion of molecules within solid-solid interfaces generates a dipole moment that changes the photovoltage and promotes charge transfer toward the surface of the electrode [88]. Furthermore, the catalysts at the solid-liquid interface can be chemically transformed as the electrolyte ions permeate into the materials. In this way, new junctions are generated, and the electrochemical properties of the system are changed during the operation of the catalytic reaction. The modulation of the solid-solid interface changes the accumulated charge concentration at the surface, the photocurrent, the photovoltage, and the onset potential for the catalytic reaction. When the various thicknesses (2–25 nm) of LaFeO_3_ are deposited on the Nb-doped SrTiO_3_, the films show a thickness-dependent charge transfer conductivity, and thus, a change in the open circuit potential. This is because the band bending of the valence-band in LaFeO_3_ at the solid-solid interface generates sensitivity to the thickness of the materials (Figure 4a) [90,91,92]. Thus, in an absorber-metal oxide device structure, the thickness of the metal oxide layer, which functions as an insulator, could control the barrier height and further improve the photovoltage [91,93,94].

The solid-solid interface can be further modulated by functionalization with various organic molecules, which induce a dipole moment, as mentioned above. In this way, the charge carrier equilibrium at the junction is engineered by tuning the surface dipole. When electron-withdrawing phosphoric acid molecules are inserted between an anchoring layer of Al_2_O_3_ and TiO_2_ in p-Si/Al_2_O_3_/TiO_2_ devices, the band level of p-Si shifts downward. As a result, the interface dipole increases the photovoltage and increases the potential energy of the charge carriers for the surface catalytic reaction. Interestingly, the surface dipole induced by the chemical interaction; in other words, the replacement of hydroxyl groups with electron-withdrawing phosphonate groups strongly affects the surface [95,96,97]. And the phenyl moieties with various substituents change the higher electron-withdrawing abilities also increase the surface generated dipole [98,99] to change the barrier height at the interface (Figure 4b). The intrinsic dipole moment of molecules that originates from substituents affects the introduced surface dipole. For example, molecules with increased magnitudes of aromaticity induce increased surface dipoles [100]. In addition, porphyrine and phthalocyanine moieties can inserted in the solid-solid junction to modulate the charge transfer at the interface [101,102]. These kinds of molecules are modulated by the various metal cation coordination and precisely tune the interface charge transfer. In this regard, the electronegativities of the substituents, the aromaticities of the organic molecules and the porphyrins and phthalocyanine moieties can modulate the charge transfer at the solid-solid interface.

The barrier height between solid-solid interface can be modulated by the surface dipole, affecting the charge transfer mechanism. When the barrier height is small, thermionic emission charge transfer is dominant, while intraband tunneling occurs when the barrier height is large. Therefore, the characteristic charge transfer distance has a reverse-volcano relationship with the surface dipole and the barrier height [99]. Insertion of organic molecules at the solid-solid interface changes the charge transfer mechanism to Z-scheme charge transfer by band bending modulation at the interface. Normally, the electrons and holes are transferred through the interface and separated in different solid materials when the band levels of two materials are staggered (Figure 4c). However, the photo-generated holes recombine with the major carrier electrons in another material at the interface in the Z-scheme mechanism, thus photo-generated electrons accumulate in the p-type materials to reduce water for production of hydrogen molecules [100]. In addition to molecular functionalization, various interface band bending methods, including the insertion of materials such as graphene, or carbon materials can change the charge transfer pathway [103,104].

The solid-solid interface can be altered during the catalytic reaction. When the electrolyte ions diffuse and permeate into the solid materials, the materials can be chemically transformed. The oxidation state of the metal cation and the crystalline structure of materials are changed as a result. Thus, a new solid-solid interface is generated during the energy converting operation, and the Fermi level of the surface changes. Because the junctions are generated and transformed continuously during the catalytic reaction, the junction is called as an adaptive junction [105,106]. The generation of metal oxide, metal hydroxide, and/or metal oxyhydroxide could alter the solid-solid interface to change the charge transfer and photovoltage [107].

We describe the solid-solid interface modulation by introducing the metal, metal oxide, and metal chalcogenides materials into the photoelectrode. The solid-solid interface could be further tuned by inserting molecules into the solid-solid interface to generate a dipole moment. As a consequence, the Fermi level at the interface can be modulated with the new charge carrier concentration equilibrium. This change leads to modulation of the band edge and band bending at the interface, affecting the charge transfer and accumulation. The changed Fermi level can simultaneously enhance the photovoltage to supply kinetic energy for the chemical reaction at the solid-liquid interface. Furthermore, a magnitude of the barrier height can determine the charge transport mechanism from thermionic emission to intraband tunneling. Although introducing multiple interfaces makes the charge transfer mechanism become more complex, it is potential strategy to effectively control charge transfer and separation.

## 3. Chemical Modulation at the Surface for Adsorption of Reactant and Catalytic Reaction

Electrocatalytic reactions typically involve multiple, successive steps occurring on the interface of a solid electrocatalyst and the electrolytes [108,109,110,111,112,113]. It is generally known that the adsorption of reactants is considered the initial step, followed by the reaction, and finally desorption for any kind of heterogeneous catalytic processes. For a targeted catalytic reaction, the optimized catalyst surface should be able to guarantee the adsorption strength is neither too strong for stabilizing the reactants and the intermediates nor too weak to prevent the poisoning of the active sites [114,115,116,117]. Due to the close and sensitive correlation between the surface properties and catalytic reactivity, fascinating achievements have been made to enhance the electrocatalytic efficiency by tuning the adsorption and underlying catalytic reactivity of the electrode materials for a specific reaction [114,118,119,120,121,122]. Specifically, the variations in the electronic states of surface atoms and geometric surface structure have been considered potential approaches to modify those two catalytic steps [119,123].

### 3.1. Modification of Intrinsic Structural Surface Geometry

#### 3.1.1. Facet Engineering

Over the past decade, it has been widely accepted that tailoring selective facet exposures can effectively improve HER/OER catalytic performances [124,125,126,127]. The fact that well-defined facets with favorable surface energies and atomic arrangements, as well as open surface structures for better reactant and intermediates adsorption ability, consequently promote the catalyzing process. The facet-specific electrocatalytic properties of electrode materials have inspired a massive number of scientific studies. As such, quantification of the facet-dependent HER/OER activity of electrocatalyst is necessary to establish a new pathway for the rational design of nanoparticle/nanocrystal catalysts.

So far, there have been intensive studies exploring novel material compounds and synthetic strategies for such purposes, which have demonstrated enhancement of real catalytic reactions. Besides the now commonplace exploration of nonnoble metal-based facet-controlled compounds, for the industrial application, reducing cost is apparently essential but remaining a huge challenge in terms of boosting activity and stability as much as compared with that of noble-based electrode materials. As such, extensive facet engineering efforts have sought to design non-precious-metal-based nanostructures for enhancement of the HER or OER by selecting the active facet to expose. Furthermore, increasing attention has been paid to high-index facets as compared to their low-index counterparts due to the high population of high-index facet active sites, including many steps, edges, and corners [124,127,128,129]. Such sites act as active centers for catalytic reactions. Therefore, the fabrication of materials with high exposure of a targeted facet is considered an efficient method for enhancing catalytic performance. For instance, controlling the crystallographic facets of the asymmetric zigzag structure of Ni_3_S_2_ (111) or of high-index faceted Ni_3_S_2_ nanosheet arrays has been successfully demonstrated to be helpful for catalyzing the HER and OER [126]. In other similar work, Feng and co-workers reported a synthetic method to grow high-index {2̅10} faceted Ni_3_S_2_ nanosheet arrays on nickel foam (NF) [130,131,132,133]. The HRTEM data obtained following the [100] crystallographic direction with the angle between the (021) and (003) plane is approximately 70.7°, which suggests that the exposed facet of the nanosheets is {2̅10} (Figure 5). Consequently, the HER and OER assessment of Ni_3_S_2_/NF with the high index {2̅10} facets indicates excellent catalytic activity. The exposure of favorable of (111) facet in MnO polyrods also was proven to be effective in enhancing HER as verified by TEM data [125].

#### 3.1.2. Heteroatom Doping

As an effective method for modifying electronic properties, heteroatom doping has gained popularity due to advantageous energy-related features to enhance the HER ascribed to the reduction of the adsorption energy barrier for the H_ad_ adsorption step [131,132,133]. Incorporation of heterogeneous elements into a host material can lead to lattice distortion of atoms due to different atomic radius, which then manipulates the redistribution of electronic density. There are mainly three types of heteroatom doping: only non-metal doping (N, S, P, etc.), only metal-doping/substituting into the main material matrix and dual doping of more than two different anionic and/or cationic elements [131,134,135,136]. For the non-metallic doping category, it has been proven that the introduction of an anion dopant plays a critical role in modulating the electronic configuration of a catalyst. In particular, the incorporation of an anion heteroatom with a different electronegativity can tailor the electronic properties of the adjacent host element, reducing the work function and improving the catalyst-reactant interaction, ultimately leading to a significant boost of catalytic performance. For example, several studies have confirmed that the importation of the more electronegative of S heteroatom as compared with that of the host can render the modulation of free energy of water molecules adsorption, dissociation and adsorbed hydrogen binding energy in the electrolyte with different pH [135]. This phenomenon was carefully demonstrated by research from Liang et al. Specifically, the authors demonstrated that S-doped MoP nanoporous fabricated using two-step chemical vapor deposition could perform very well in a pH-universal electrolyte system with a very low overpotential of 86 mV at a current density of 10 mA · cm^−2^ and a low Tafel slope of 34 mV · dec^−1^ in acidic media. Another source of catalytic improvement has also been proposed to be the interaction of catalysts surface and electrolyte was also enhanced due to the presence of S. Experiments and theoretical simulations of properties such as band structure and density of states (DOS) have confirmed that the absorption of protons is favored during the HER process when using S-doping. Recently, our group reported the effectiveness of heteroatom nitrogen doping in the MXene matrix using facile calcination under an ammonia atmosphere accompanied with adjusting the temperature treatment [137]. We employed X-ray photoelectron spectroscopy as the main analysis tool to investigate the role of the N dopant in inducing a change to the electronic configuration of the catalyst by different N-related species. In addition, DFT calculations also supported experimental data that N contained components in N-Ti_3_C_2_T_x,_ including Ti-N, N-H, and O-Ti-N, all contributed to a significant improvement of observed HER electrocatalytic activity. Specifically, at the optimum temperature (600 °C), N-Ti_3_C_2_T_x_@600 electrocatalysts required an overpotential as low as 198 mV at a current density of 10 mA · cm^−2^ and a small Tafel slope of 92 mV dec^−1^ in acidic media (Figure 6). Interestingly, a new concept of orbital modulation deriving favorable water adsorption and dissociation for the HER under alkaline conditions has been reported by Qian’s group [138]. Specifically, their careful structural characterizations indicated that the developed carbon-doped molybdenum disulfide adjusted the electronic properties and coordination of the parent MoS_2_ phase, thereby accelerating the reactant adsorption capability and reaction kinetics. Additionally, the authors also found that the tuning orbital orientation was responsible for the exceptional alkaline HER of modified molybdenum disulfide. Up to now, recent progress in anion-doped catalysts has logged many interesting achievements regarding synergistic effects that stem from two or more anions within the electrocatalyst, which enabled better performance than a single non-metal dopant [139,140].

Although it is challenging to prepare due to the usually high energy required, the positive effects of metallic dopant on HER/OER have been well established in previous literature. Employing many advanced techniques such as synchrotron-based X-ray absorption near-edge structure, X-ray photoelectron spectroscopy, auger electron spectroscopy, ultraviolet photoemission spectroscopy, and density functional theory calculations, it is consistently revealed that the improved performance of catalytic water oxidation/reduction is attributed to the changes to the catalyst’s electronic structure and the downshift of the d-band center after metal doping, which then favors the Gibb free energy for proton adsorption. As an specific example for this class, researchers demonstrated that metal (Ni, Mn, Fe) doping can result in a change in electronic structure of the catalyst and downshift the d-band center for accelerating the HER with an outstanding performance at a low cell voltage (1.43 V @ J10) after doping in a CoP matrix [141]. Other work reported the use of Ir-doped NiV-layered double hydroxide, in which Ir-O-V plays the role of an active site for water adsorption and dissociation [134]. In addition, the author also found out that the existence of such active groups can simultaneously boost HER and OER by reducing the charge density on adjacent oxygen and increasing the charge density on V atoms, respectively (Figure 7a,b). This work provided guidance for designing various transition metal-doped compounds such as oxides, sulfides, and phosphides for energy-related electrocatalysis applications. Meanwhile, the cation dopant inducing the changes in the valence of the metal has been demonstrated in work from Yamauchi’s laboratory [134]. In detail, they synthesized composite IrO_x_-TiO_2_-Ti (ITOT) catalysts, which exhibited excellent OER performance with 1.43 V vs. RHE at J10. This material also exhibited good stability 100 h after a chronopotentiometric test under acidic solution. Upon extensive experimental and theoretical investigations, they concluded that their method enabled control of the Ir-valence to achieve high OH intermediate concentration at the catalyst surface, which is crucial for obtaining outstanding catalytic OER performance. In 2017, Deng and coworkers introduced their studies about the multiscale structural, and electronic control of MoS_2_ foam consisted of plenty of mesopores, vertically aligned two-dimensional layers, and cobalt atoms dopant [131]. Through careful experiments and calculations, they proved that the obtained-materials exhibited significantly promoted HER activity (Figure 7c–f). The metallic dopant as the atomic-scale engineering played a role in modulation the H adsorption on MoS_2_ to a suitable degree and stabilized the structure to boost the HER reactivity.

#### 3.1.3. Vacancy Engineering

Besides the abovementioned strategies, defect engineering has been shown that it can provide the potential method to generate active sites for nanomaterials for the electrocatalytic process through the effective redistribution of the local charges or creating of the additional active center of an adjacent atom near the local defects. Among many types of defects, the vacancy has proven to be an effective surface defect and been investigated in various kinds of electrocatalysts [142,143,144,145]. Therefore, within the scope of this article, we concentrate on the discussion about its contribution in the electrocatalytic process. The recent intriguing progress has demonstrated that defect optimized heterocatalysts can enable the substantially improved activity for HER/OER, implying the possibility of the practical implementation of this strategy. As is well known, introducing vacancies into electrode materials can create low coordinated sites located near the vacancy sites, and the corresponding induced dangling bonds can act as adsorption positions for reactants and mediate gas-evolving reactions [145]. In addition, the creation of a vacancy resulted from the introduction of dopant can also cause the charge compensation effect, thereby boosting the activation of oxygen-related species, which are imperative in redox reactions. For example, through a comprehensive study of the kinetics for water dissociation on a distorted spinel Ni-O-O layer on a MoS_2_ plate, the authors found that the HER and OER activity was enhanced 40 and 2.5 times, respectively, as compared to bare spinel nickel cobaltite (Figure 8) [125]. The DFT calculations results indicated that oxygen vacancy-mediated Ni^3+^ sites were responsible for the OER improvement, which was due to the decrease of the activation energy barrier for water dissociation that is critical for promoting HER activity [125]. Another example is the synthesis of VS_2_ nanosheets with the rich defect features for significant catalytic HER improvement through a simple solvothermal synthetic method [143]. Zhang et al. revealed that the high density of defects generated on the interlayer-expanded surface of VS_2_ successfully modified the electronic structure of VS_2_ to achieve optimal free energy for hydrogen adsorption. In other words, the adsorption kinetics of the reactant has been significantly boosted, leading to outstanding electrocatalytic activity.

Additionally, the oxygen vacancies mediated generation of coordinatively unsaturated cations are considered as highly active centers in oxidation reaction at the metal/oxide interface. A number of published papers have shown that these active oxygen vacant sites surpass reactivity as compared with that of the oxygen sites with a higher coordination number, which is located in the lattice oxygen of the metal oxides. Thoughtful theoretical simulations evidently revealed that the adsorption strength of reactant and intermediates adsorption could be adjusted to suit the specific reductive or oxidative catalytic reaction [142,144].

### 3.2. Introduction of Surface Functionalities

#### Controlling the Coordination Number of Exposed Atoms

The effect of particle size on electrocatalytic activity has long been an attractive topic, and there are many different explanations of the underlying mechanism for improving catalyst performance. Mostly, researchers have argued that the small size of heterogeneous nanoparticle catalysts enables an increase in reactivity. Previous literature has shown that such improvements are strongly correlated with the adsorption of reactants and intermediates of a specific reaction to edges and surface-exposed sites that exist in high density on small nanoparticles. These sites have surface energies that are easily accessible to the reactant with much lower Gibb free energy as compared to the other sites, thereby promoting catalytic activity [27,146,147].

Very recently, single-atom catalysts (SACs) have attracted considerable attention in the field of heterogeneous electrocatalysts due to their unique structural properties with the highly distributed metal element at the atomic level [1,27]. In other words, following what is understood from the size-dependent effect, researchers expected that when decreasing the nanoparticle to the atomic level, one could expect the best mass activity. Thus far, fascinating work has been conducted regarding SACs. Along with conventional methods, the fast development of many advanced characterization techniques, such as X-ray absorption spectroscopy, synchrotron-based atomic X-ray absorption fine structure (EXAFS) spectroscopy, has enabled the clear observation and demonstration of SACs. Thanks to the well-defined, isolated atoms on the substrate, the maximum atomic usage efficiency can be achieved [148,149]. For example, single atomic catalysts of Ni atoms uniformly distributed on a supported N-doped matrix has been prepared and confirmed to be efficient for the OER. This SACs has a well-controlled coordination geometry with effective electronic coupling via the Ni-N coordination that moves the Fermi level down and lowers the adsorption energy of the intermediates, thus facilitating OER kinetics [150]. Apart from the geometric effect of exposing a large population of single active sites in the SACs, tuning of the relative metal-support interaction (RMSI) also confers advantages to the outstanding photo-derived H_2_ production, as reported by Guo’s group [133]. Using various material characterization techniques as well as simulation studies, they found that the strong RMSI of single Pt atoms on g-C_3_N_4_ supports resulted in rich N vacancies can enhance the performance of HER through the successful capture of electrons and beneficial geometric effects (Figure 9). They also reported the PH_3_-promoted method to prepare phosphorus-coordinated single metal atoms on g-C_3_N_4_ nanosheets via Lewis acid interaction between the constituents. Experiments and simulations suggested that the Pt SACs on g-C_3_N_4_ affords novel electronic-rich features that are conceptually different from those of the well-known system of a single atom with an electronic deficient state. These novel features resulted in new electronic properties, which are preferable for the adsorption of oxygen-related intermediates. This effect is responsible for the boosting of HER photocatalytic H_2_ production by a factor of four compared to that of the state-of-the-art N-coordinated PdSAs embedded on g-C_3_N_4_ nanosheets. It was also noted that this concept could be applied to other noble metals such as Ru and Rh.

## 4. Conclusions

We provided an understanding of the charge transfer and separation processes in addition to the chemical reactant absorptions behaviors that are helpful for enhancing the efficiency of conversion in photo/electrocatalysis water oxidation and reduction. Engineering the solid-liquid and solid-solid interfaces can control the charge carrier dynamics and the catalyzing reaction on the surface of the catalysts. The surface structure, including surface defects, atomic arrangements, coordination states, and surface termination can affect the electronic and chemical properties for the charge accumulation and chemical reaction kinetics on the surface. Furthermore, the surface properties can be modulated through passivation, insertion of materials, and functionalization with organic molecules. Specifically, surface modulation tailors the surface dipole moment and band bending to control electrochemical properties, such as barrier height and photovoltage. As a result, the surface chemical structure can also influence the chemical potentials of the elements on the surface, adsorption energies of the chemical reactants, and the Gibbs free energy of the intermediate states during the electrocatalytic process.

Although surface modulation plays an important role both in charge carrier dynamics and the kinetics of the chemical reaction, combining the absorber and catalysts into one system requires additional consideration of issues including band alignment, balanced charge carrier extraction rates, and stabilities against the operational conditions. In the below perspective, we propose additional considerations for the future fabrication of photo/electrodes using the rational design of efficient photoconversion systems.

(1) Harsh conditions, such as acidic or alkaline electrolyte solutions. For water oxidation and reduction, degrades photoelectrode efficiency. Previously, passivation layers were introduced on the surface of the absorber to overcome this stability issue. However, the passivation layer also covers the catalytically active sites where the reactants adsorb and react to generate the fuels at the solid-liquid interface. Thus, the passivation layer degrades the efficiency of photo energy conversion into chemical energy [151,152,153]. Therefore, the introduction of stable catalysts or surface modulation of the passivation layer by incorporation of substitution of elements, doping, and functionalization is needed.

(2) The introduction of multiple interfaces makes the charge transfer and separation process complex. This sometimes decreases the photovoltage for the catalytic reaction. However, the multi-junctions are important for maximizing light absorption in a broad region and enhancing photoconversion and catalytic efficiency. As a consequence, various efforts have been devoted to developing tandem-cells and integrated photovoltaic-photo/electrochemical cells for water splitting. Understanding and integrating the cells by the precise selection of materials and modulation of the interface is needed to minimize the charge transfer resistance and further optimize the over-potentials for the catalytic reaction.

(3) Balanced charge consumption rates for the HER and/or OER should be considered. The kinetic mechanism of OER is more sluggish compared to that of the HER. This slow hole carrier consumption would also retard the other half-reaction due to the recombination of generated carriers with remaining charge carriers. Thus, some additives such as electron/hole scavengers are could be introduced into the system, or the precise control of band alignments for efficient charge transfer to the reactants could be one of the solvation methods.

(4) The morphology of the surface is controlled to increase the surface area to adsorb more reactants. However, the morphology of the catalysts affects how the interface works for charge transfer/separation into reactants. For example, if the size of the catalysts is too small for them to interact with each other on the photoelectrode, so the electrochemical properties of the system are determined by the mixed solid-solid and solid-liquid interface, which is known as a pinch-off effect [154]. Thus, the morphology of the catalyst on the surface should be considered in terms of both reactant diffusion and charge carrier equilibrium with the electrolyte system.

## Figures and Tables

**Figure 1 molecules-25-01965-f001:**
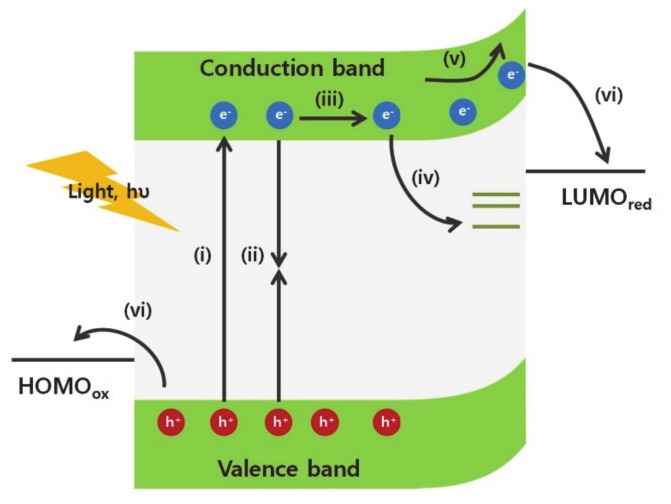
Illustration of photogenerated charge carriers dynamics in photocathodic catalysts (i) excitation of electrons and holes by absorption of light above band gap of semiconductors, (ii) radiative recombination, (iii) diffusion of charge carriers, (iv) charge carriers trapped in the surface defect states, (v) drifted flow of electrons induced by band bending on the surface, and (vi) charge separation into the electron/hole acceptors at the interface and chemical reaction of electron/hole acceptors.

**Figure 2 molecules-25-01965-f002:**
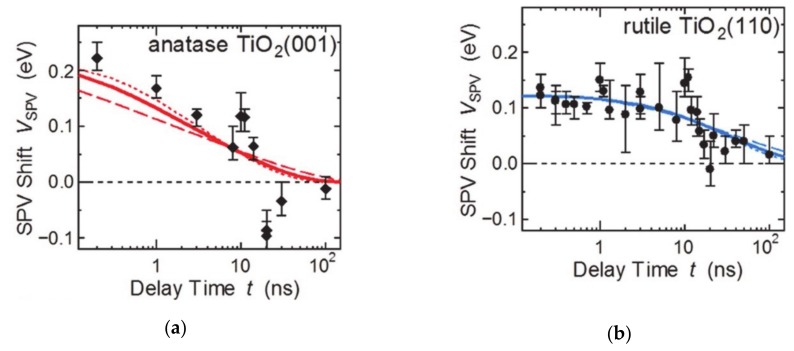
Surface photovoltage analysis for facet-dependent charge carrier relaxation time difference according to the barrier height at the surface. The figure is reproduced from Ref. [44] with permission from the Royal Society of Chemistry.

**Figure 3 molecules-25-01965-f003:**
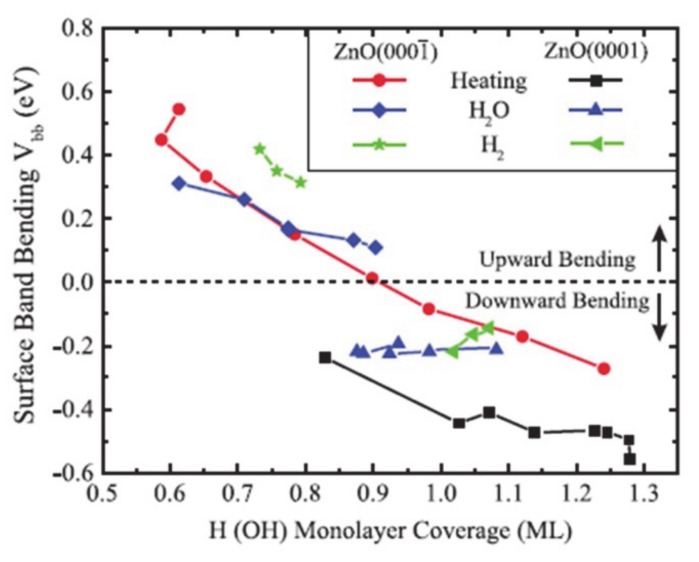
The surface termination of the ZnO changes the band bending at the surface, which is observed by the X-ray photoelectron spectroscopy. The figure is reproduced from Ref. [46] with permission from the American Physical Society.

**Figure 4 molecules-25-01965-f004:**
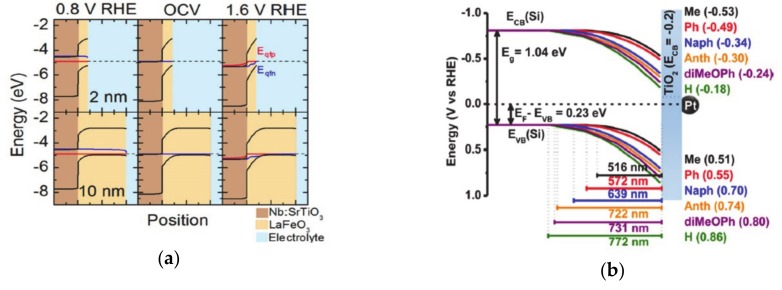
(**a**) The band bending introduced by perovskite oxide on the surface, which is reproduced from Ref. [90] with permission from the Royal Society of Chemistry. The thickness of perovskite oxide changes the Fermi level at the surface and, as a result, changes the open-circuit potentials. (**b**) The band bending of the semiconductor functionalized with molecules, which have various dipole moments, as reproduced from Ref. [100] with permission from the Royal Society of Chemistry (**c**) The solid-solid interface band alignments are changed by the introduction of organic molecules within solid-solid interfaces. The figure is reproduced from Ref. [98] with permission from the Royal Society of Chemistry.

**Figure 5 molecules-25-01965-f005:**
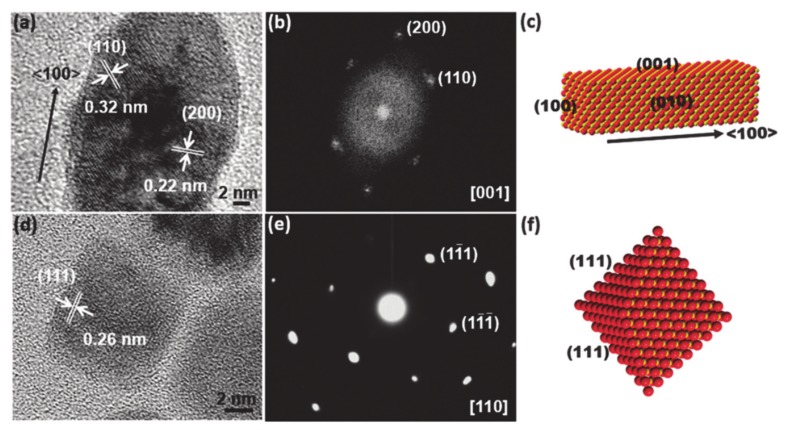
(**a,b**) The high-resolution transmission electron microscope (TEM) image of a single rod of MnO polypods (**a**) and the corresponding selected area electron diffraction (SAED) pattern (**b**). (**c**) Schematic illustration indicating the growth direction and exposed planes of MnO nanorods on MnO polypods. (**d,e**) The high-resolution TEM image of a single MnO octahedral nanoparticle (NP) (d) and the corresponding SAED pattern (**e**). (**f**) Schematic illustration indicating the growth direction of MnO octahedral NPs. O is red and Mn is yellow. Reproduced with permission [125]. Copyright 2015, Royal Society of Chemistry.

**Figure 6 molecules-25-01965-f006:**
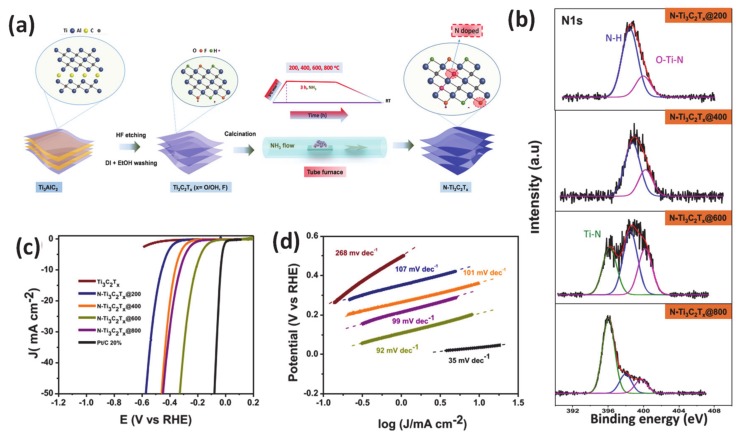
(**a**) Illustration of synthesis of nitrogen-doped MXene (N-Ti_3_C_2_T_x_) from the Ti_3_AlC_2_ MAX Phase, (**b**) core-level X-ray photoelectron spectroscopy (XPS) of the N 1s of N-doped MXene samples annealed at various temperatures, (**c**) HER polarization curves of various obtained-electrocatalysts, (**d**) corresponding Tafel plots. Reproduced with permission [137]. Copyright 2019, American Chemical Society.

**Figure 7 molecules-25-01965-f007:**
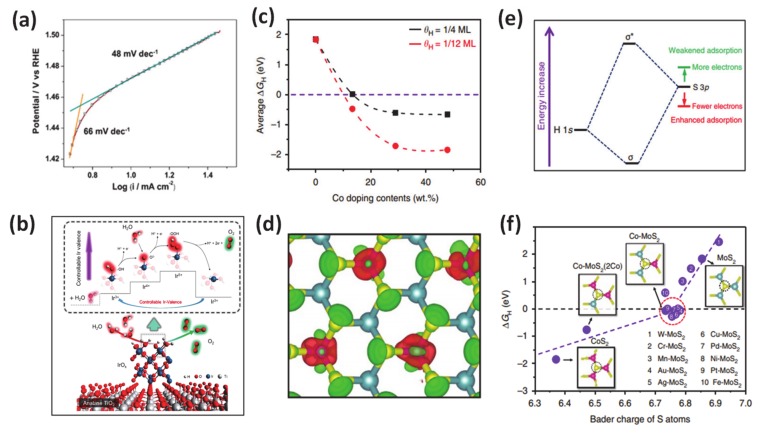
(**a**) Oxygen evolution reactions (OER) activity: Tafel plot of the IrO_x_-TiO_2_-Ti (ITOT) catalyst at the scan speed of 10 mV/s. (**b**) Proposed OER mechanism of ITOT catalyst. Theoretical calculations for the effect of Co doping contents on HER of MoS_2_. Reproduced with permission [134]. Copyright, 2019, American Chemical Society. (**c**) Average hydrogen adsorption free energy ∆G_H_ on S atoms versus the Co doping contents, considering different coverage of 1/4 monolayer (ML) and 1/12 ML. (**e**) Schematic diagram of the bonding of H 1s orbital and S 3p orbital (from MoS_2_), where depletion of electrons on S atoms will lower the orbital position and enhance the H-S bond. (**d**) The differential charge density of Co-doped MoS_2_ (Co doping content of 13.3 wt%, Co: Mo atomic ratio of 1:2). Red and green contours represent electron accumulation and depletion, respectively. The isosurface level is set to be 0.11 e/Bohr3. (**f**) ∆G_H_ on S atoms versus the Bader charge of S atoms for different structures. The insets are the atomic configurations of one S atom bonding with three Co, two Co, and one Mo, one Co, and two Mo, as well as three Mo atoms, respectively. Green balls: Mo; yellow balls: S; pink balls: Co. Reproduced with permission [131]. Copyright 2017, Springer Nature.

**Figure 8 molecules-25-01965-f008:**
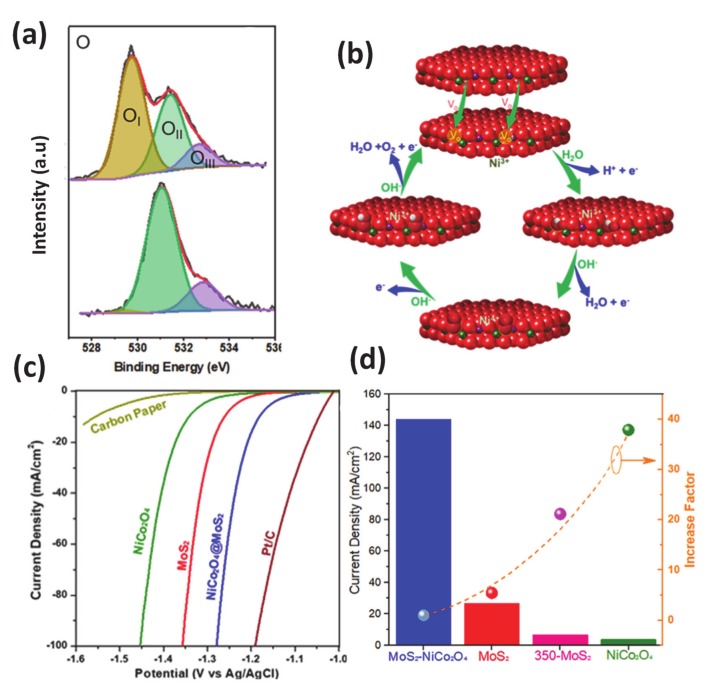
(**a**) XPS spectra of O 2s (**b**) The proposed OER cycles over the NiCo_2_O_4_@MoS_2_, where the formation of OH^−^, O^2−^, and OOH^−^ intermediates are reconciled with oxygen vacancy-mediated Ni^3+^ center; the spinel structure with metal ions between interstices of two oxygen layers is used in this cartoon. HER performances of MoS_2_, NiCo_2_O_4_, and NiCo_2_O_4_@MoS_2_ samples: (**c**) Polarization curves in 1M NaOH at room temperature referred to Pt/C and carbon paper and (**d**) current density with an overpotential n = 0.3 V vs. RHE (column bars), and the current density increase factor calculated by comparing j_NiCo2O4@MoS2_/J_sample_. Produced with permission [125]. Copyright 2019, American Chemical Society.

**Figure 9 molecules-25-01965-f009:**
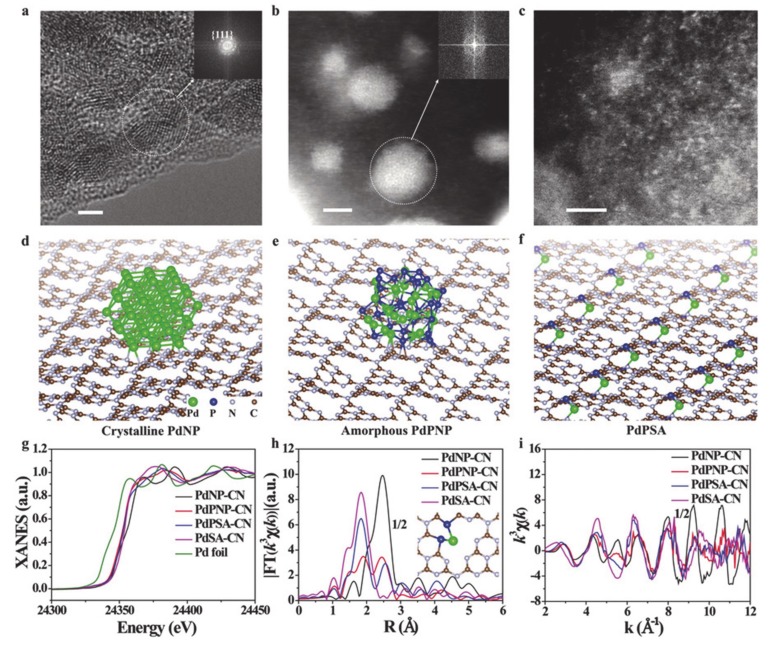
**(a**) High resolution transmission electron microscopy (HRTEM) image of PdNP-CN. High angle annular dark field scanning transmission electron microscope (HAADF-STEM) images of (**b**) PdPNP-CN and (**c**) PdPSA-CN. The geometry structures of (**d**) PdNP-CN, (**e**) PdPNP-CN, and (**f**) PdPSA-CN. (**g**) Pd K-edge X-ray absorption near edge structure (XANES) spectra and the corresponding k3-weighted Fourier transform (FT) spectra at (**h**) R and (**i**) k space. Scale bar: 2 nm. Reproduced with permission [133]. Copyright 2019, John Wiley and Sons.

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
