# Peer review of "Understanding Surface Modulation to Improve the Photo/Electrocatalysts for Water Oxidation/Reduction"

_molecules, 2020, doi:10.3390/molecules25081965_

Round 1

Reviewer 1 Report

This comprehensive review from Lee and coworkers discusses the role of interfacial chemistry and structure in photo and electrocatalysts for water oxidation and reduction. There is quite a bit of content in this review and it often reads like a tutorial, which I think will be of great use to many student readers or newcomers to the field. 

One minor concern that may be worth addressing in the introduction - there is some discussion of band bending and its role in charge transfer. How does this apply to nanoscale catalytic materials especially at the smaller size scale? Some additional elaboration here could be helpful.

One additional (very recent) reference that may benefit the completeness of the present review, perhaps in section 3.2 (for example), is "Designing Nanoparticle Interfaces for Inner-Sphere Catalysis". This relates more to the electrocatalyst emphasis of this review, but I think it makes many of the same points about the important of role of interfaces in these processes. https://pubs.rsc.org/en/content/articlelanding/2020/dt/d0dt00785d#!divAbstract

There are a few minor grammatical/stylistic mistakes that would benefit from revision. One more careful reading and edit should serve the manuscript well. 

Reviewer 2 Report

Presented review is focused on the description of the methodologies of the surface modification of the nanomaterials considered as catalysts for the reduction or the oxidation of water. The work constitutes a well-written and designed knowledge compendium concerning the mentioned research topic. The whole is given in an understandable manner and supported by the specific schemes and may definitely contribute to the increase in knowledge in the described research area. Paper is valuable but some minor improvements are suggested by the reviewer regarding mainly the extending of some too generally given aspects. All details are listed below.

  • In section 1.2. Authors mentioned that HER performance is strongly sensitive to the pH of the electrolyte applied. This pH sensitivity needs to be discussed in more detail.
  • Next, it was stated that the catalytic reaction requires the modification of catalysts to control the complex physical and chemical reactions in the photocatalysts. Some examples of such modification should be provided by Authors.
  • In section 2.1. Authors reported that the incorporation of metal onto the surface is an effective way to control the overpotentials for the reaction. In paper, few examples of metals that may be incorporated should be given.
  • Few other methods applied for the regulation of the defects of the surface apart from the mentioned introducing passivation layers or annealing should be indicated by Authors (section 2.1.).
  • Authors mentioned in section 2.3. that the solid-solid interface can be modulated by the functionalization with various organic molecules – some specific examples of such organic molecules need also to be given in the paper.

Reviewer 3 Report

A well written review. I am very positive in publishing it and I only have a minor comment. In the introduction it is said that H2, O2, CH4, and alcohol are valuable fuels. However, I don't think O2 can be regarded as a fuel. The authors need to remove O2 from the list or explain what they mean.
